# Physicochemical and Rheological Properties of Succinoglycan Overproduced by *Sinorhizobium meliloti* 1021 Mutant

**DOI:** 10.3390/polym16020244

**Published:** 2024-01-15

**Authors:** Jaeyul Kim, Jae-pil Jeong, Yohan Kim, Seunho Jung

**Affiliations:** 1Department of Bioscience and Biotechnology, Microbial Carbohydrate Resource Bank (MCRB), Konkuk University, 120 Neungdong-ro, Gwangjin-gu, Seoul 05029, Republic of Korea; jincairu@naver.com (J.K.); bruce171525@gmail.com (J.-p.J.); shsks1@hanmail.net (Y.K.); 2Department of System Biotechnology, Microbial Carbohydrate Resource Bank (MCRB), Konkuk University, 120 Neungdong-ro, Gwangjin-gu, Seoul 05029, Republic of Korea

**Keywords:** succinoglycan, NTG, overproducing mutant, physiochemical, rheological, thermal stability

## Abstract

Commercial bacterial exopolysaccharide (EPS) applications have been gaining interest; therefore, strains that provide higher yields are required for industrial-scale processes. Succinoglycan (SG) is a type of bacterial anionic exopolysaccharide produced by *Rhizobium*, *Agrobacterium*, and other soil bacterial species. SG has been widely used as a pharmaceutical, cosmetic, and food additive based on its properties as a thickener, texture enhancer, emulsifier, stabilizer, and gelling agent. An SG-overproducing mutant strain (SMC1) was developed from *Sinorhizobium meliloti* 1021 through N-methyl-N′-nitro-N-nitrosoguanidine (NTG) mutation, and the physicochemical and rheological properties of SMC1-SG were analyzed. SMC1 produced (22.3 g/L) 3.65-fold more SG than did the wild type. Succinoglycan (SMC1-SG) overproduced by SMC1 was structurally characterized by FT-IR and ^1^H NMR spectroscopy. The molecular weights of SG and SMC1-SG were 4.20 × 10^5^ and 4.80 × 10^5^ Da, respectively, as determined by GPC. Based on DSC and TGA, SMC1-SG exhibited a higher endothermic peak (90.9 °C) than that of SG (77.2 °C). Storage modulus (G′) and loss modulus (G″) measurements during heating and cooling showed that SMC1-SG had improved thermal behavior compared to that of SG, with intersections at 74.9 °C and 72.0 °C, respectively. The SMC1-SG′s viscosity reduction pattern was maintained even at high temperatures (65 °C). Gelation by metal cations was observed in Fe^3+^ and Cr^3+^ solutions for both SG and SMC1-SG. Antibacterial activities of SG and SMC1-SG against *Escherichia coli* and *Staphylococcus aureus* were also observed. Therefore, like SG, SMC1-SG may be a potential biomaterial for pharmaceutical, cosmetic, and food industries.

## 1. Introduction

Natural polymers have been extensively studied as biomaterials in various fields such as pharmaceuticals, cosmetics, and food [1]. Recently, the industrial use of microbial exopolysaccharide (EPS), a natural polymer, has attracted considerable attention [2]. Microbial-derived EPSs are attractive biomaterials that have received scientific and commercial attention for decades. Additionally, water-soluble microbial polysaccharides have advantages such as high water-binding capacity, creamy texture, and excellent viscosity for food-related products [3]. Industrially used microbial EPSs include hyaluronic acid, alginate, xanthan, gellan, curdlan, succinoglycan, and cellulose [4,5]. EPS is an environmentally friendly, biocompatible, and biodegradable polymer with significant potential to provide a wide range of rheological properties, such as thickening, stabilizing, texturizing, emulsifying, and gelling [6]. Some microbial polysaccharides also have beneficial health effects, including blood cholesterol lowering, anticancer, antibacterial, and immunostimulatory properties [7,8].

Succinoglycan (SG) is a microbial exopolysaccharide produced by soil bacteria such as *Rhizobium* and *Agrobacterium* species [9,10]. The SG produced by *Sinorhizobium meliloti* 1021 consists of seven glucose molecules and one galactose with non-carbohydrate substituents, including succinyl, acetyl, and pyruvyl groups [11,12]. The type and content of these substituents vary depending on the source of the *rhizobium* or culture conditions [13,14]. SG solutions exhibit a characteristically high viscosity owing to the presence of succinyl groups, which have high potential as water-soluble thickeners [15]. Previous studies have shown that the viscosity of SG increases with the degree of succinyl group substitution [15]. Additionally, SG demonstrates high stability even under extreme operating conditions, such as high temperatures and pressures, extreme salinity and pH, and high shear rates [16,17]. These properties make SG suitable for applications in the cosmetics, food, and pharmaceutical industries as a thickener, gelling agent, stabilizer, texturizer, and emulsifier [18]. Additionally, recent studies have reported that SG exhibits antibacterial activity, attributed to the activation of the mitogen-activated protein kinase MAPK/interleukin IL-6 pathway, the mechanism that changes during *Listeria* infection, affecting processes related to protein synthesis, glycolysis, and oxidative stress [19].

Recently, many studies have been conducted to increase the yield of microbial polysaccharides such as hyaluronic acid, gellan, xanthan, pullulan, levan, and alginate [20,21,22,23,24]. For the industrial use of microbial polysaccharides, low-cost and high-efficiency production is required. Therefore, methods such as the development of strains that overproduce microbial polysaccharides, optimization of medium conditions, and extraction processes are being studied [25,26,27,28]. Previous research showed that the ExoR and ExoS proteins of *Sinorhizobim meliloti* 1021 regulate SG production [29]. Generally, NTG mutagenesis has been used for many purposes, including changes in the structure, activity, function, and yield of EPS [30,31,32,33]. We used a previously reported method to increase EPS yield without any significant changes in the structure, activity, or function of succinoglycan (SG). In this study, we developed an SG-overproducing strain using an NTG mutation and analyzed the physiochemical and rheological properties of SG overproduced by the strain.

Previous studies have shown that the yield of polysaccharides produced by microorganisms is increased using N-methyl-N′-nitro-N-nitrosoguanidine (NTG) mutagenesis [34,35]. In this study, a mutant strain (SMC1) that overproduces SG was selected from *Sinorhizobium meliloti* 1021. Given that the SG (SMC1-SG) produced by SMC1 had different yields depending on the mannitol concentration used as a carbon source in the medium, the production medium was optimized according to the mannitol content in the medium. SG and SMC1-SG structures were compared through Fourier transform infrared (FT-IR) and nuclear magnetic resonance (NMR) spectroscopic analysis. To evaluate thermostability, thermal analyses such as differential scanning calorimetry (DSC) and thermogravimetric analysis (TGA) were performed. The molecular weight and non-carbohydrate substituent content were analyzed by gel permeation chromatography (GPC) and NMR peak analysis.

Viscosity was measured under various concentrations, salinities, pH, and temperature conditions, such measurements being important from an industrial application perspective. Studies related to the different rheological properties of SG and SMC1-SG over-concentration, temperature range, pH, and salinity conditions will help in broadening their application scope in the cosmetic and food industries. Previous studies have demonstrated the potential application of metal-cation-mediated crosslinking hydrogels using anionic polysaccharides [36,37]. Gelation induced by SMC1-SG was investigated for several metal cations. Additionally, to evaluate SMC1-SG for use as a biomaterial, its antibacterial activity against *Escherichia coli* and *Staphylococcus aureus* was investigated. These results suggest that SMC1-SG can be used as a biomaterial in the food, cosmetics, and pharmaceutical industries.

## 2. Materials and Methods

### 2.1. Materials

The bacterial strain (*Sinorhizobium meliloti* 1021) was supplied by the Microbial Carbohydrate Resource Bank (MCRB) at Konkuk University (Seoul, Republic of Korea). Mannitol, glutamic acid, potassium phosphate monobasic, potassium phosphate dibasic, magnesium sulfate, and calcium chloride were purchased from Daejung (Busan, Republic of Korea). N-methyl-N′-nitro-N-nitrosoguanidine, sodium thiosulfate, Tris-Maleate, xanthan, alginate, and pullulan were purchased from Sigma-Aldrich (Steinheim, Germany).

### 2.2. Isolation of Succinoglycan (SG)

*Sinorhizobium meliloti* 1021 was cultured for 2 days at 30 °C and 200 rpm in seed medium containing 0.1% trace elements. The composition of the seed medium was mannitol 5 g/L; glutamic acid 1 g/L; dibasic potassium phosphate 1 g/L; magnesium sulfate 0.2 g/L; calcium chloride 0.04 g/L. After subculturing two or more times, the cells were cultured in a production medium at 30 °C and 200 rpm for 7 days. The composition of the production medium was mannitol 10 g/L; glutamic acid 1 g/L; potassium phosphate dibasic 10 g/L; potassium phosphate monobasic 10 g/L; magnesium sulfate 0.2 g/L; calcium chloride is 0.04 g/L. The cultured medium was centrifuged at 8000 rpm for 15 min at 4 °C to obtain the supernatant. After concentrating the supernatant to 1/5 volume using an evaporator, SG was precipitated by adding a volume of ethanol thrice that of the concentrated supernatant. The amount of SG was measured after the precipitate was dried in an oven at 60 °C for 24 h to remove any ethanol residue. The SG and SMC1-SG yields were optimized and measured by adjusting the mannitol content.

### 2.3. N-Methyl-N′nitro-N-nitrosoguanidine (NTG) Mutagenesis

*Sinorhizobium meliloti* 1021 was cultured for 2 days and centrifuged at 1000 rpm for 3 min. The supernatant was removed, and the cells were treated twice with a 20 mM solution of initiator Tris-Maleate buffer at pH 8.5 [38]. The NTG solution was dissolved at 0.5 mg/mL in 10% acetone and treated at 30 °C and 200 rpm for 30 min [34]. The cells were treated twice with sodium thiosulfate in saline water. The cell solutions were diluted 1/10^6^ and cultured in a seed medium containing 1.5% agar at 30 °C for 2 days. Colonies that were larger and more vivid were selected, and the mutant strain that produced the highest amount of SG was named SMC1 (Figure 1).

### 2.4. Fourier Transform Infrared (FT-IR) Spectroscopy

The FT-IR spectrum was measured with an FT-IR spectrometer in ATR mode (0.5 cm^−1^ resolution, Bruker, Germany) at wavenumbers of 4000–700 cm^−1^.

### 2.5. ^1^H Nuclear Magnetic Resonance (NMR) Spectroscopy

The ^1^H NMR spectrum was obtained with a 600 MHz Bruker Avance spectrometer (Bruker, Karlsruhe, Germany). The sample solutions comprised 7.5 mg SG in 750 μL deuterated water (D_2_O, 99.95%) and were measured at 25 °C.

### 2.6. Differential Scanning Calorimetry (DSC)

DSC was performed using a Discovery DSC 2500 (TA Instruments, New Castle, TA, USA). Dried samples (10 mg) were placed in a sealed aluminum pan and heated under N_2_. Heat flows were recorded at a scan rate of 10 °C/min over a temperature range of 25–200 °C.

### 2.7. Thermal Gravimetric Analysis (TGA)

TGA was performed using a Discovery TGA 5500 (TA Instruments, USA). The 10 mg samples were prepared in a dried state and the weight loss was observed at a rate of 10 °C/min over 20–600 °C.

### 2.8. Molecular Weight by Gel Permeation Chromatography (GPC)

GPC was performed using a Waters 2414 refractive index detector and Waters 1525 Binary pump with H_2_O. Sodium nitrate (0.02 N) and pullulan were used as the elution solvent and calibration standard, respectively. The flow rate was set at 0.8 mL/min.

### 2.9. Rheological Measurements

The viscoelasticity properties were evaluated using a DHR-2 rheometer (TA-Instruments, USA). The sample solutions were prepared with an SG weight percentage of 1.0%. The angular frequency was swept with a fixed strain of 0.5% in the 0.1–100 rad/s range. The amplitude sweep strain was measured in the range from 0.1% to 1000% at 1 Hz. The storage modulus (G′) and loss modulus (G″) of the sample solution were measured under various pH conditions (pH 3–11), 10% salt aqueous solutions (NaCl, CaCl_2_, FeCl_2_, MgCl_2_), and concentrations (0.5–1.5%). The intersections of G′ and G″ with temperature during heating and cooling were also measured.

### 2.10. Gelation Test by Metal Cations

In the gelation test by metal cations, Fe^3+^, Cr^3+^, Sr^2+^, Zn^2+^, Al^3+^, Co^2+^, Cu^2+^, Ca^2+^, Mn^2+^, Pb^2+^, and Fe^2+^ solutions were used, along with the control (distilled water). SG (10 mg) was stirred into 1 mL of 15 mM metal solution in a glass vial. After 24 h, the vial was turned over and gelation was observed.

### 2.11. Antibacterial Test

*E. coli* and *S. aureus* were used to evaluate antibacterial activity. An amount of 1 mL of 1 wt% SG solution was added to an 8 mm paper disk, placed in the center of LB agar containing bacterial liquid, and incubated at 37 °C for 24 h. Additionally, 1 mL of 1 wt% polysaccharide solution was added to 5 mL of the bacterial suspension. After incubation at 37 °C for 24 h, 200 μL of the bacterial suspension was retrieved and measured with a UV–vis spectrophotometer (UV2450, Shimadzu Corporation, Kyoto, Japan) at 600 nm. The antibacterial effects were calculated using the following equation:$$\mathrm{Antibacterial}\;\mathrm{effect}\;(\%)=\frac{I_{1}-I_{2}}{I_{1}}\times 100$$

*I*_1_ and *I*_2_ are the OD values of the control group (bacterial suspension) and experimental group (bacterial suspension with polysaccharide solution), respectively. All measurements were performed in triplicate.

## 3. Results and Discussion

### 3.1. Optimization of SG Yield

The growth of *S. meliloti* 1021 depends on several factors, such as culture time, aeration, agitation, pH, inoculation volume, and media source [39,40]. Microbial growth affects the yield of microbial EPS, and the carbon source of the medium from which the most SG was produced by *Sinorhizobium meliloti* 1021 was mannitol [41]. Carbon source is an important factor in the polysaccharide yield by microorganisms [42]. SMC1-SG yield was investigated based on the mannitol content in the production medium.

#### SG Yield Change Depending on Mannitol Content in Medium

SG yield by the wild-type strain was 6.72 g/L. SG production changes depending on the type and content of the carbon source [41,43]. The maximum yield of SG produced by the wild type of *Sinorhizobium meliloti* 1021 was 7.8 g/L when the mannitol content was 50 g/L [9]. SMC1-SG yields were optimized by increasing the mannitol content of the production medium from 5 g/L to 50 g/L. SMC1-SG yields were compared by increasing the amount of mannitol. Increasing from 5 g/L to 10 g/L and 20 g/L significantly increased the yield, whereas at 30 g/L, the yield decreased (Figure 2). The highest yield, 22.3 g/L, was obtained when the mannitol content was 10 g/L, which was a 3.65 times increase compared to the wild type (control).

### 3.2. Characterization of SG

SG produced by *S. meliloti* 1021 consists of seven glucose molecules and one galactose [44]. The non-carbohydrate groups include pyruvyl, acetyl, and succinyl groups [45] (Figure 3). To directly determine the structure of an isolated SG, it is necessary to characterize it. The characteristics of SG produced from wild-type and SMC1-SG produced from SMC1 were investigated through various spectrometric analyses, such as FT-IR, NMR, DSC, TGA, and GPC. SMC1-SG produced by the NTG mutant strain SMC1 showed no changes in structure, activity, or functions, according to FT-IR, ^1^H NMR peak analysis, and GPC data. However, SMC1-SG showed higher thermostability than that of SG, which could be explained by the high succinylation of SMC1-SG [46].

#### 3.2.1. Fourier Transform Infrared (FT-IR)

Figure 4a and Table 1 show the FT-IR spectra and functional groups of SG and SMC1-SG, showing that the characteristic peaks are the same. The absorption peak at approximately 3350 cm^−1^ and the band observed at 2893 cm^−1^ correspond to the stretching vibration of -OH and the asymmetric vibrations of CH_2_ and -CH_3_ of the polysaccharide, respectively [27]. The signal at approximately 1723 cm^−1^ was the C=O stretching vibration of SG, whereas the signals at 1609 cm^−1^ and 1371 cm^−1^ were attributed to asymmetric and symmetric COO^-^ vibrations, respectively [47]. These signals corresponded to the succinyl, pyruvyl, and acetyl functional groups. Additionally, the peak at 1070 cm^−1^ indicates the presence of C-O-C, and the absorption peak at 890 cm^−1^ indicates that SG contains β-glycosidic bonds [44]. The FT-IR results are consistent with those of previous studies on succinoglycan [17,25,34].

#### 3.2.2. Nuclear Magnetic Resonance (NMR) Spectroscopic Analysis

SG and SMC1-SG were analyzed using ^1^H NMR to obtain additional information about the substituents in these samples (Figure 4b). The double peak assigned to the succinyl methylene proton of the SG sample and SMC1-SG represents 2.50 ppm [14]. In both samples, acetyl groups were observed at 2.01 ppm, and methyl groups of pyruvate were found at 1.44 ppm [48]. When the NMR peak intensity of the pyruvate signal was set to 1.00, the signal intensities for all other functional group substitutions were similar (Table 2). These signals corresponded to those of previous studies showing ^1^H NMR of SG [34,48].

#### 3.2.3. Differential Scanning Calorimetry (DSC)

The thermal behavior of SG and SMC1-SG was measured by DSC (Figure 5a). In thermal analysis, SG used as a control showed an endothermic peak at 77.22 °C, but SMC1-SG showed an endothermic peak at a higher temperature, 90.94 °C. As the freeze-dried sample was analyzed by DSC, endothermic peaks, including residual moisture, hydrogen bonding, and electrostatic interaction, appeared more clearly at both temperatures [49]. Because the appearance of endothermic peaks upon heating is generally a result of dehydration, structural collapse, and hydrogen bond destruction [50], SMC1-SG with relatively higher M_w_ (Table 3) showed improved thermal stability [51].

#### 3.2.4. Thermal Gravimetric Analysis (TGA)

TGA revealed weight loss with increasing temperature (Figure 5b). The lyophilized samples exhibited two stages of weight loss. In the first stage, weight losses of 14.22% and 11.74% occurred for SG and SMC1-SG, respectively, in the 80 °C region. These are mostly weight losses due to dehydration, and approximately 90% of SMC1-SG maintained its physical integrity and stability across temperatures [52]. Additionally, a slight weight loss observed at approximately 200 °C for SMC1-SG can be attributed to the further vaporization of bound water [53].

In the second step, the SG and SMC1-SG weight losses in the 300 °C region were 41.9% and 40.7%, respectively. Weight losses in this region indicate the thermal destruction of the structure formed by the cross-linking and strong bonds of polysaccharides [54]. The curves were almost identical until decomposition at 300 °C. Considering the remaining weight percentage (%), SMC1-SG showed a slightly enhanced thermal stability [27,55]. SG and SMC1-SG also have more residual weight than polysaccharides such as xanthan and starch [56].

#### 3.2.5. Gel Permeation Chromatography (GPC)

GPC was performed to determine the molecular weights of SG and SMC1-SG. SG had a number average (M_n_) of 3.33 × 10^5^ Da and a weight average (M_w_) of 4.20 × 10^5^ Da. SMC1-SG had an M_n_ of 4.49 × 10^5^ Da and M_w_ of 4.80 × 10^5^ Da, which were higher than those of SG. Given that SG is composed of repeating octameric oligosaccharide units with one to two succinyl groups, higher succinylation may indicate that SG contains more succinylated repeating units [45]. This suggests that SMC1-SG has slightly higher succinylated repeating units. The polydispersity index was 1.26 and 1.07 for SG and SMC1-SG, respectively.

### 3.3. Rheological Test of SG

Maintaining the viscosity of polysaccharide solutions under various conditions is beneficial for a wide range of applications [17]. Current commercially applied thickeners maintain their viscosity under a variety of conditions [57]. Therefore, the viscosities of SG and SMC1-SG were measured under various concentrations, salinities, pH, and temperature conditions. Shear thinning has also been observed in fluids with significant hydrogen bonding, such as shear polymer solutions, paints, motor oils, molecular liquids, liquid polymers, and glasses [58,59,60]. Xanthan gum, a biopolymer, is commonly used to form shear-thin fluids [61]. Given that SG, an acidic polysaccharide with carboxylic groups and primary and secondary alcohol groups, has multiple H-bonds, it showed typical shear thinning, also known as pseudoplasticity, which could be characterized by a decreasing apparent viscosity with increasing shear rate. The intersection of G′ and G″ during heating and cooling indicated the thermal stability of SG and SMC1-SG.

#### 3.3.1. Angular Frequency Test

Frequency sweep tests were performed on the SG and SMC1-SG solutions (Figure 6a). Sample solutions were prepared at 1 weight percent. The storage modulus (G′) and loss modulus (G″) of SG and SMC1-SG increased as the angular frequency (rad/s) increased, and the two moduli were almost parallel. At low frequencies, the SG and SMC1-SG chains were entangled. On the other hand, at high frequencies, the chains were disentangled. This implied that SG and SMC1-SG solutions exhibited elastic properties in the low-frequency region and viscous properties in the high-frequency region. All measurements were conducted in triplicate (Table 3 and Table 4).

#### 3.3.2. Oscillation Test

The change in G′ with strain for 1 weight percent SG and SMC1-SG solutions is shown (Figure 6b). The term “viscoelastic behavior” refers to the fact that polymer fluids can behave similarly to elastic solids but behave like viscous liquids, which are dependent on stress [62]. The linear viscoelastic region (LVR) of a material is defined as the linear relationship between the composite stress and oscillatory strain [63]. Therefore, the large LVR region, i.e., maintaining the G′ value under significant stress, means that SG and SMC1-SG solutions have viscoelastic behavior [64]. All measurements were conducted in triplicate (Table 5 and Table 6). Experiments were performed at 25 °C.

#### 3.3.3. Viscosity Measurement according to Concentration and Temperature

Viscosity was measured as a function of the concentration of SG and SMC1-SG and shear rate versus concentration (Figure 7a,b). The viscosity of SG as a function of shear rate was higher at 1.50% and 1.25% concentrations compared to SMC1-SG. At the lowest concentration (0.50%), SMC1-SG presented higher viscosity than that of SG, with an increasing shear rate. However, the viscosity reduction patterns of SG and SMC1-SG were similar at different concentrations. The results showed that the viscosity was directly proportional to the concentration of SG and SMC1-SG and that the solution exhibited pseudoplastic behavior with increasing shear rate. The shear rate is directly proportional to the concentration [65].

The effect of temperature on viscosity was measured (Figure 8a,b). The viscosity of 1 weight percent SG and SMC1-SG solutions was stable at 25 °C, 35 °C, 45 °C, and 55 °C. The viscosity reduction pattern of the SG solution decreased sharply at 65 °C, but that of the SMC1-SG solution was maintained. This indicates the thermal stability of SMC1-SG. At high temperatures, common EPSs change to an irregular state due to heat, and their viscosity decreases [66].

#### 3.3.4. Measurement of G′ and G″ during Heating and Cooling

The G′ and G′ intersections of SG and SMC1-SG aqueous solutions versus temperature during heating and cooling are shown in Figure 9a,b. Both the heating and cooling rates were 10 °C/min. The gel and sol properties change at the point where G′ and G″ intersect [67]. These changes are caused by heat-induced disturbances. At the start of heating, the G′ of the SG solution had no temperature dependence and then dropped at approximately 60 °C and intersected with G″ at 71.9 °C. Upon cooling, G′ gradually increased and crossed G″ at 66.9 °C. In terms of SMC1-SG, the crossover temperature was 74.9 °C when heated and 72.0 °C when cooled, which were higher than those of SG. This structural transition of SG confirms that it is a flexible chain or coil [66]. Figure 9c shows the sigmoidal patterns of G′ and G″ during heating and cooling, showing the difference in tan δ (G″/G′). During cooling, there was a rapid sigmoidal increase in G′, which shifted to a progressively lower temperature; however, upon heating, the decrease in G′ shifted in the opposite direction, resulting in progressively larger thermal hysteresis [68].

#### 3.3.5. Viscosity Measurement according to pH and Salt Condition

The SMC1 viscosity of SG was measured over a wide pH range (Figure 10a,b). The viscosity of polysaccharides decreases as pH changes from 7.0 [47]. SG and SMC1-SG maintained the viscosity reduction patterns at various pH values.

Viscosity was measured under various salt conditions such as calcium, magnesium, potassium, and sodium (Figure 11a,b). The viscosity reduction patterns of SG are constant under salt conditions [69]. Both SG and SMC1-SG were stable under all salt conditions tested. In industrial processes, products with high viscosities are very important for applications.

### 3.4. Application Capabilities through Gelation and Antibacterial Test

Metal-cation-induced gelation and antibacterial activity were investigated to evaluate applicability in various fields. Recent studies have reported that EPS generally exhibits significant antibacterial effects [70].

#### 3.4.1. Metal Chelating Gelation by Metal Cation

The gelation of SG and SMC1-SG was observed using various metal-cation-reducing agents (Figure 12). Metal cations act as gelling agents or cross-linkers that bind to anionic polysaccharides. Alginate hydrogels with metal cross-linkers such as Ca^2+^ have been studied for biomedical applications [71,72]. In previous SG hydrogel studies, 15 mM Fe^3+^ solution was used as a cross-linker. [73]. In this study, gelation was observed in various 15 mM metal solutions. SG and SMC1-SG were observed to gel with various metal cation solutions (15 mM), such as Fe^3+^ Cr^3+^, Fe^2+^, Ca^2+^, Co^2+^ Pb^2+^, Al^3+^, Cu^2+^, and Mn^2+^. For both SG and SMC1-SG, gelation was observed in Fe^3+^ and Cr^3+^ solutions.

#### 3.4.2. Antibacterial Test

The antibacterial properties of SG and SMC1-SG were evaluated against *E. coli* and *S. aureus*. SG and SMC1-SG showed the ability to inhibit the growth of *E. coli* and *S. aureus* on agar medium (Figure 13a). The SG inhibition zone diameters for *E. coli* and *S. aureus* were up to 18.7 ± 1.8 mm and 13.4 ± 0.2 mm, respectively. For SMC1-SG, *E. coli* and *S. aureus* showed 15.1 ± 1.6 mm and 14.2 ± 0.6 mm, respectively. As shown in Figure 13b,c, additional antibacterial experiments were performed on microbial-derived polysaccharides SG, SMC1-SG, xanthan, alginate, and pullulan. Antibacterial activity was quantified by measuring the optical density. SG demonstrated a bactericidal activity of 84.49 ± 1.56% against *E. coli* and 76.82 ± 1.21% against *S. aureus*. SMC1-SG presented a bactericidal capacity of 73.31 ± 1.24% against *E. coli* and 80.14 ± 1.48% against *S. aureus*. This indicates that SG and SMC1-SG effectively inhibited bacterial growth. This is because of succinyl substituents, which create a low-pH environment, chelate metal ions, and use their surfactant ability to inhibit bacterial cell growth [74].

## 4. Conclusions

We developed the SG-overproducing mutant strain, SMC1, using an NTG mutation in *Sinorhizobium meliloti* 1021. The structure, thermal stability, rheology, cation-induced gelation, and antibacterial properties of SG produced by SMC1 were investigated. NMR, FTIR, and GPC analyses revealed that SG produced by SMC1 had a similar structure to SG produced by the wild-type strain but showed improved thermal stability based on DSC and TGA results compared to that of SG. The rheological tests revealed that SG and SMC1 maintained their viscosity reduction patterns at various pH values, salinities, and temperatures. In particular, SMC1-SG was able to maintain viscosity reduction patterns even at high temperatures compared to SG. Both SG and SMC1-SG effectively inhibited bacterial growth. Therefore, SMC1-SG has high thermal stability, viscosity at high temperatures, antibacterial activity, and gelation by some trivalent cations, suggesting its potential as a biomaterial with industrial scope in various fields, such as food science, cosmetics, and biotechnology, that require microbial polysaccharides.

## Figures and Tables

**Figure 1 polymers-16-00244-f001:**
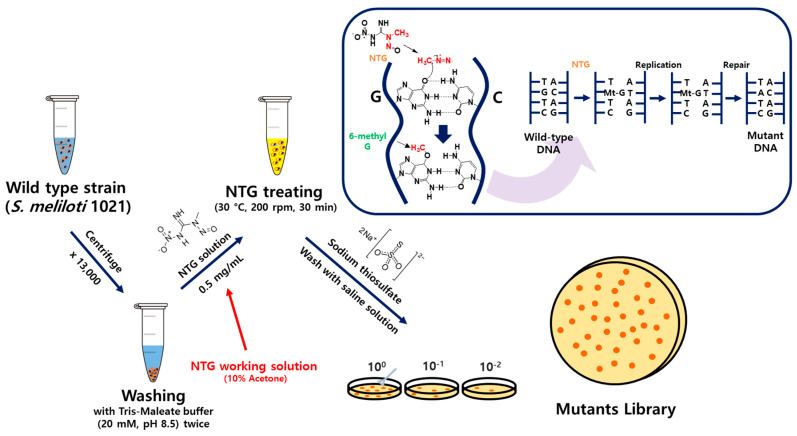
NTG mutagenesis protocol.

**Figure 2 polymers-16-00244-f002:**
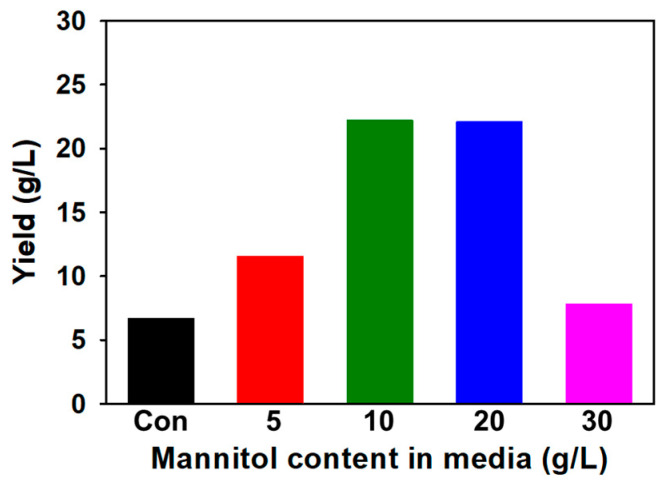
The yield of succinoglycan produced by SMC1 according to mannitol content in media.

**Figure 3 polymers-16-00244-f003:**
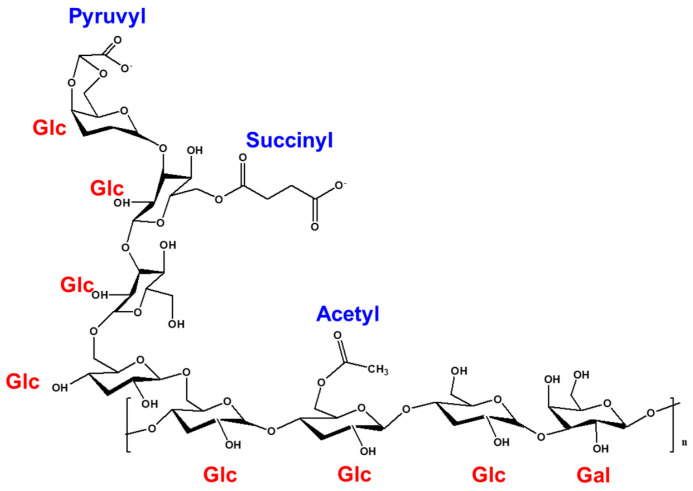
Structure of succinoglycan.

**Figure 4 polymers-16-00244-f004:**
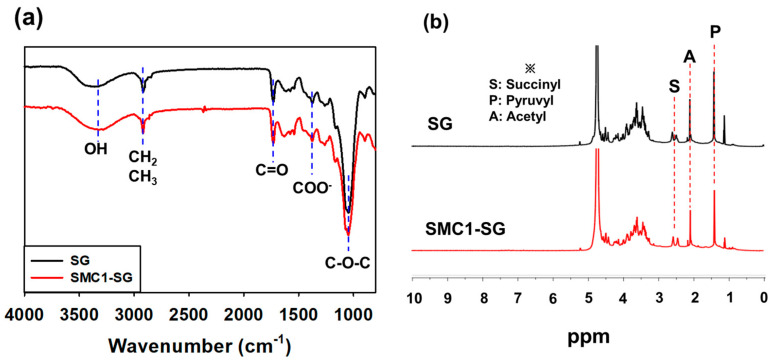
(**a**) FT-IR and (**b**) ^1^H NMR of SG and SMC1-SG.

**Figure 5 polymers-16-00244-f005:**
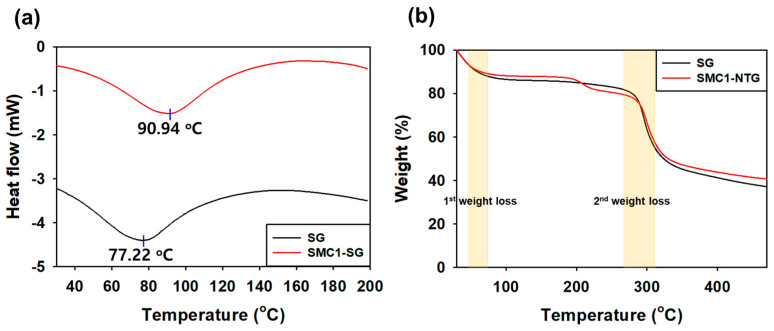
(**a**) DSC and (**b**) TGA curves of SG and SMC1-SG.

**Figure 6 polymers-16-00244-f006:**
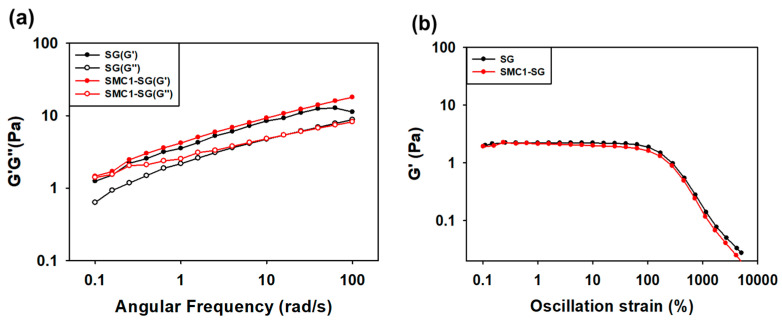
Elastic modulus G′ and viscosity modulus G″ for SG and SMC1-SG solutions in (**a**) angular frequency and (**b**) oscillation strain measurements.

**Figure 7 polymers-16-00244-f007:**
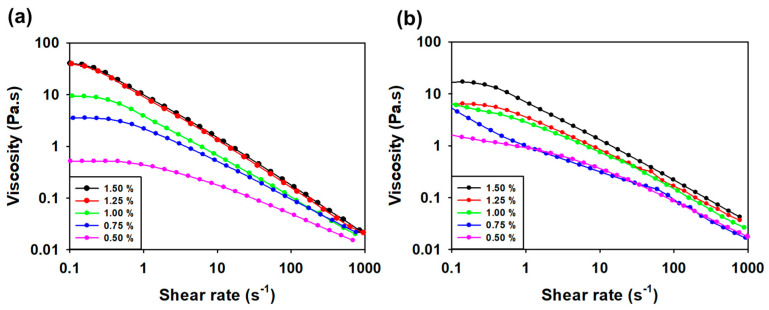
Dependence of viscosity on the shear rate for different concentrations of (**a**) SG and (**b**) SMC1-SG.

**Figure 8 polymers-16-00244-f008:**
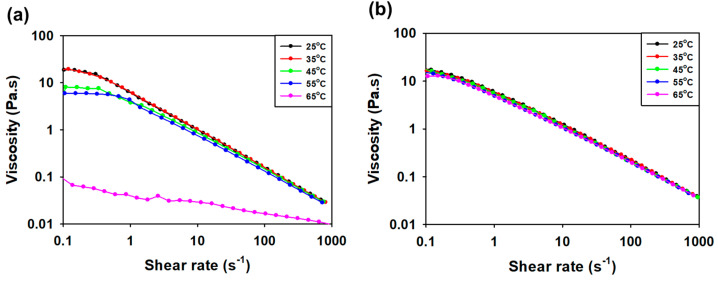
Influence of temperature on viscosity of 1 weight percent (**a**) SG and (**b**) SMC1-SG solution.

**Figure 9 polymers-16-00244-f009:**
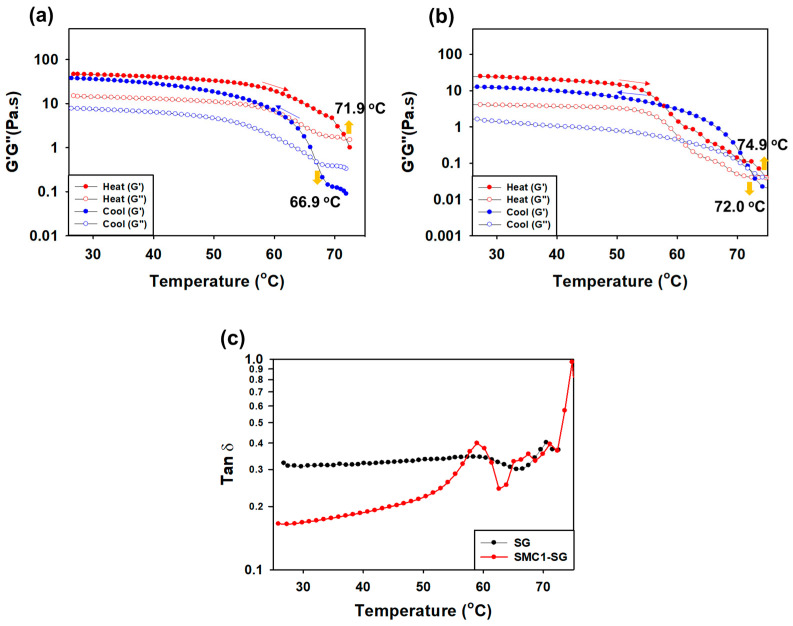
Elastic modulus G′ and viscosity modulus G″ for 1 weight percent solutions during heating and cooling: (**a**) SG, (**b**) SMC1-SG, and (**c**) tan δ.

**Figure 10 polymers-16-00244-f010:**
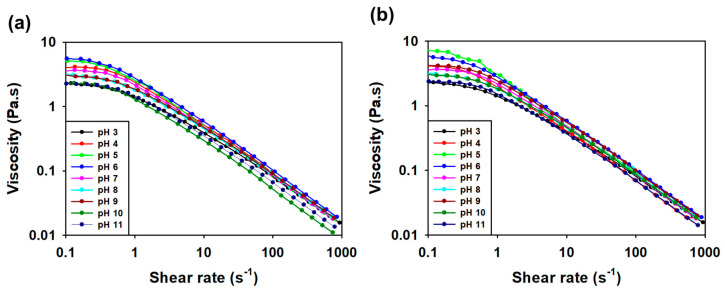
Effect of pH on 1 weight percent (**a**) SG and (**b**) SMC1-SG aqueous solutions in the pH range of 3.0–11.0.

**Figure 11 polymers-16-00244-f011:**
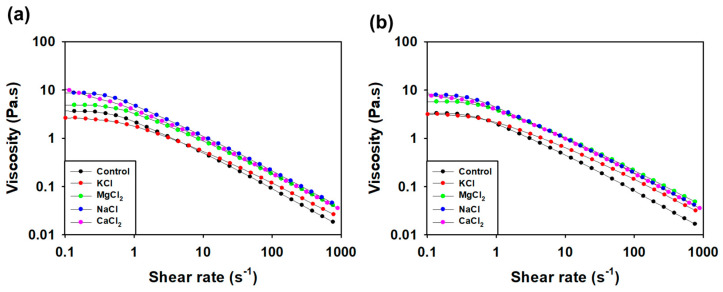
Effect of salts on 1 weight percent (**a**) SG and (**b**) SMC1-SG aqueous solutions in 10% KCl, MgCl_2_, NaCl, and CaCl_2_ conditions.

**Figure 12 polymers-16-00244-f012:**
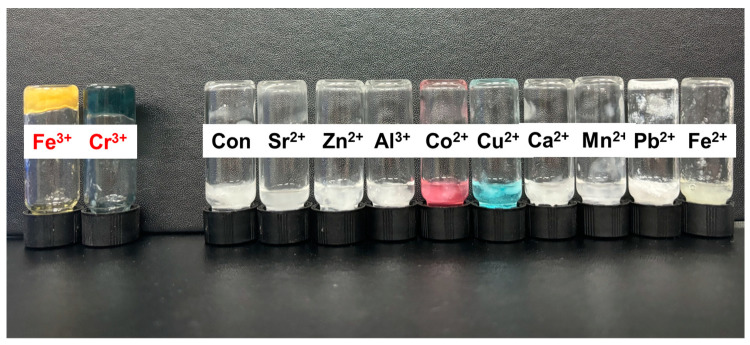
Gelation of SMC1-SG by Fe^3+^ and Cr^3+^.

**Figure 13 polymers-16-00244-f013:**
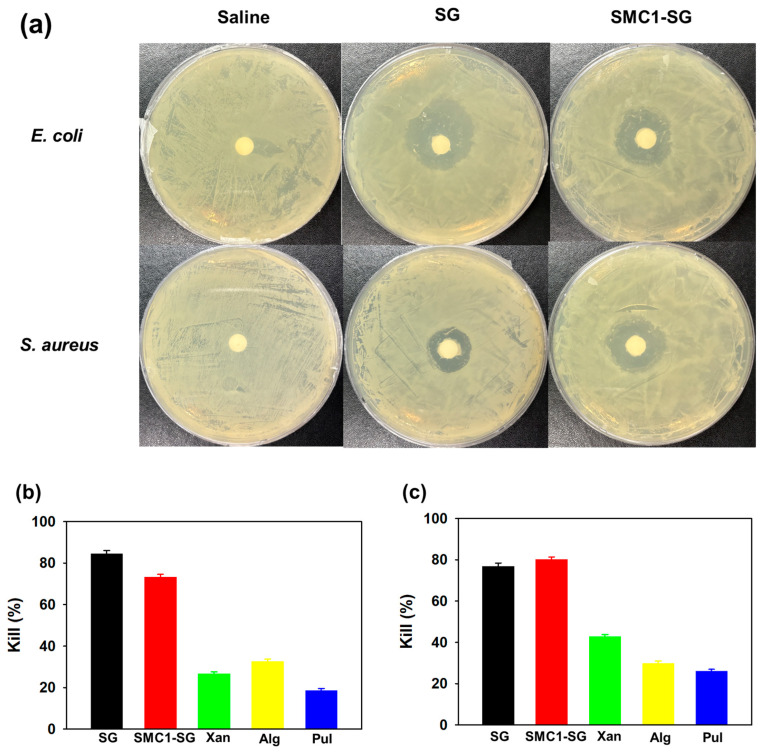
(**a**) Inhibition areas of SG and SMC1-SG against *E. coli* and *S. aureus* were observed on agar plates. Evaluation of the antibacterial activity against (**b**) *E. coli* and (**c**) *S. aureus* of SG, SMC1-SG, xanthan, alginate, and pullulan.

**Table 1 polymers-16-00244-t001:** FT-IR peak ratio analysis.

Functional Group	Wavenumber (cm^−1^)	Characteristics
-OH	3350	Broad stretching vibration
-CH_2_, -CH_3_	2893	Stretching vibration
C=O	1723	Stretching vibration
Asymmetric COO^−^	1609	Asymmetric stretching vibration
Symmetric COO^−^	1371	Symmetric stretching vibration
C-O-C	1070	Stretching vibration
β-glycosidic bond	890	Stretching vibration

**Table 2 polymers-16-00244-t002:** ^1^H NMR signal intensities ratio analysis.

Sample	Succinyl	Acetyl	Pyruvyl
SG	1.28	0.85	1.00
SMC1-SG	1.34	0.89	1.00

**Table 3 polymers-16-00244-t003:** Rheological measurements of SG for angular frequency.

Angular Frequency (rad/s)	G′ (Pa)	G″ (Pa)	Tan (δ)	Step Time (s)
0.1	1.245 ± 0.022	0.632 ± 0.019	0.508 ± 0.021	65.848
1	3.542 ± 0.008	2.168 ± 0.014	0.612 ± 0.011	180.618
10	8.442 ± 0.006	4.722 ± 0.025	0.559 ± 0.012	214.077
100	11.235 ± 0.192	8.805 ± 0.076	0.783 ± 0.144	246.779

**Table 4 polymers-16-00244-t004:** Rheological measurements of SMC1-SG for angular frequency.

Angular Frequency (rad/s)	G′ (Pa)	G″ (Pa)	Tan (δ)	Step Time (s)
0.1	1.460 ± 0.092	1.401 ± 0.137	0.959 ± 0.108	65.812
1	4.183 ± 0.043	2.529 ± 0.031	0.604 ± 0.037	180.396
10	9.269 ± 0.017	4.800 ± 0.172	0.517 ± 0.122	213.774
100	17.934 ± 0.061	8.212 ± 0.265	0.457 ± 0.142	247.009

**Table 5 polymers-16-00244-t005:** Rheological measurements of SG for oscillation strain.

Oscillation Strain (%)	G′ (Pa)	G″ (Pa)	Tan δ	Step Time (s)
0.1	2.006 ± 0.165	1.732 ± 0.021	0.863 ± 0.048	9.392
1	2.205 ± 0.113	1.932 ± 0.034	0.876 ± 0.042	58.582
10	2.203 ± 0.071	1.831 ± 0.028	0.831 ± 0.044	97.875
100	1.858 ± 0.068	2.012 ± 0.042	1.082 ± 0.04	147.119
1000	0.139 ± 0.001	0.260 ± 0.002	1.870 ± 0.001	245.546

**Table 6 polymers-16-00244-t006:** Rheological measurements of SMC1-SG for oscillation strain.

Oscillation Strain (%)	G′ (Pa)	G″ (Pa)	Tan δ	Step Time (s)
0.1	1.920 ± 0.126	1.373 ± 0.011	0.715 ± 0.042	9.401
1	2.126 ± 0.141	1.393 ± 0.018	0.655 ± 0.057	58.689
10	1.984 ± 0.092	1.383 ± 0.021	0.697 ± 0.034	98.077
100	1.608 ± 0.051	1.254 ± 0.032	0.779 ± 0.037	147.27
1000	0.116 ± 0.002	0.432 ± 0.005	3.703 ± 0.002	245.663

## Data Availability

Data is contained within the article.

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
