# Peer review of "Physicochemical and Rheological Properties of Succinoglycan Overproduced by Sinorhizobium meliloti 1021 Mutant"

_polymers, 2024, doi:10.3390/polym16020244_

Round 1

Reviewer 1 Report (Previous Reviewer 2)

Comments and Suggestions for Authors

In this manuscript, a succinoglycan overproducing mutant strain was developed, and its structural, physicochemical, and rheological properties were investigated. SMC1-SG showed enhanced thermal stability and viscosity maintenance at high temperatures.  The antibacterial activity and cation-induced gelation were also studied. The author claims its potential as a biomaterial for pharmaceutical, cosmetic, and food industries. 

The author has incorporated revisions and corrections in response to my earlier feedback. The scientific clarity has been enhanced. Overall, the content is insightful; however, there is a need for language refinement to enhance clarity and improve understanding. Consider revising the text for better articulation and coherence to ensure that readers can grasp the concepts more effectively.

For example: Line 361-364, “Previous studies show that the viscosity of food hydrocolloid SG is constant under saline conditions [66]. Both SG and SMC-SG were stable in all saline conditions tested. Previous research shows that the viscosity of SG decreases with increasing ionic strength [63].” The sentences lack both logical connection and contextual coherence.

Comments on the Quality of English Language

Certain aspects of the English writing remain challenging to follow. Consequently, I recommend a comprehensive review of the writing for further improvement.

Author Response

Reviewer 2 Report (New Reviewer)

Comments and Suggestions for Authors

see PDF file attached.

Comments on the Quality of English Language

English writing is fine, while some abbreviations and Latin name of microbes should be more clearly presented in the manuscript.

Author Response

This manuscript is a resubmission of an earlier submission. The following is a list of the peer review reports and author responses from that submission.

Round 1

Reviewer 1 Report

Comments and Suggestions for Authors

The authors carried out a large number of experiments but in the paper they only quickly summarize the results without a discussion and/or interpretation and/or  comparison with literature data. Thus, it is impossible to understand both the  aim of the paper and the conclusions that can be drawn from this study. At this stage, therefore, in my opinion the article is not ready for publication.

Comments on the Quality of English Language

poor

Reviewer 2 Report

Comments and Suggestions for Authors

In this manuscript, a succinoglycan overproducing mutant strain was developed, and its structural, physicochemical, and rheological properties were investigated. SMC1-SG showed enhanced thermal stability and viscosity maintenance at high temperatures.  The antibacterial activity and cation-induced gelation were also studied. The author claims its potential as a biomaterial for pharmaceutical, cosmetic, and food industries. 

Comments:

1.     The FTIR and NMR spectra for SG and SMC1-SG show no discernible differences. An explanation for this observation is needed.

2.     In Figure 5b, there is a slight weight loss observed for SMC1-SG (labeled as SMC1-NTG), which should be explained. Additionally, the figure caption for the red line is marked as "SMC1-NTG." The author should address these anomalies for clarity.

3.     In Section 3.3.2, the statement "the linear viscoelastic regions of SG and SMC1-SG solutions were between 200% and 250% strain" appears inaccurate. Clarification or correction is needed to accurately define the linear viscoelastic regions.

4.     It is advisable for the rheology measurements to be conducted multiple times, with statistical results summarized. This would enhance the reliability and robustness of the reported data.

5.     The manuscript lacks an explanation for the observed shear thinning mentioned in lines 256-257. It would be valuable for the author to elucidate the potential causes, such as H-bond disruption.

6.     Figure 7 indicates a higher viscosity for SG compared to SMC1-SG at 1.5% and 1.25%. The author should provide an explanation for this unexpected observation to ensure a thorough understanding of the results.

Comments on the Quality of English Language

Language can be understood.